

# COVID-19 and regional differences in the timeliness of hip-fracture surgery: an interrupted time-series analysis

Davide Golinelli[1], Jacopo Lenzi[1], Emanuele Adorno[1], Maria Michela Gianino[2] and Maria Pia Fantini[1]

[1] Department of Biomedical and Neuromotor Sciences, University of Bologna, Bologna, Italy
[2] Department of Public Health Sciences and Pediatrics, University of Turin, Turin, Italy

## ABSTRACT

**Background**. It is of great importance to examine the impact of the healthcare reorganization adopted to confront the COVID-19 pandemic on the quality of care provided to non-COVID-19 patients. The aim of this study is to assess the impact of the COVID-19 national lockdown (March 9, 2020) on the quality of care provided to patients with hip fracture (HF) in Piedmont and Emilia-Romagna, two large regions of northern Italy severely hit by the pandemic.

**Methods**. We calculated the percentage of HF patients undergoing surgery within 2 days of hospital admission. An interrupted time-series analysis was performed on weekly data from December 11, 2019 to June 9, 2020 ($\approx$6 months), interrupting the series in the 2nd week of March. The same data observed the year before were included as a control time series with no "intervention" (lockdown) in the middle of the observation period.

**Results**. Before the lockdown, 2-day surgery was 69.9% in Piedmont and 79.2% in Emilia-Romagna; after the lockdown, these proportions were equal to 69.8% (−0.1%) and 69.3% (−9.9%), respectively. While Piedmont did not experience any drop in the amount of surgery, Emilia-Romagna exhibited a significant decline at a weekly rate of −1.29% (95% CI [−1.71 to −0.88]). Divergent trend patterns in the two study regions reflect local differences in pandemic timing as well as in healthcare services capacity, management, and emergency preparedness.

## INTRODUCTION

Worldwide, hip fracture (HF) represents an important public health concern that determines relevant functional impairments in the individuals who experience it, especially the elderly (*Tedesco et al., 2018*; *Svedbom et al., 2013*). Due to the increasing incidence of osteoporosis, the global number of HFs will reach approximately 8.2 million in 2050 (*Sambrook & Cooper, 2006*). In Italy, HFs accounted for 99,103 hospitalizations, 1,122,714 occupant days and 77,543 surgical procedures in 2018.

The growing burden that HF causes on healthcare systems also has to do with the intensive use of healthcare resources required by this condition (*Beaupre et al., 2019*;

Corresponding author
Davide Golinelli,
davide.golinelli@unibo.it

*Haentjens et al., 2010*). HF patients represent a particularly challenging population (*Hansen et al., 2013*; *Moja et al., 2012*), due to the high post-operative mortality rate caused by surgery, functional impairment, and limited mobility (*Beaupre et al., 2019*; *Haentjens et al., 2010*; *Braithwaite, Col & Wong, 2003*).

Timely surgery within 48 h of hospital admission for HF is a well-established strategy that leads to better functional outcomes and lower mortality rates (*Moja et al., 2012*; *Carretta et al., 2011*; *Shiga, Wajima & Ohe, 2008*). The UK National Institute for Health and Care Excellence (NICE) and other international guidelines indicate 48 h as the ideal time to operate on a patient with HF. Scientific evidence points out that the earlier the surgery, the better the outcomes in terms of mortality, complications, length of hospital stay, time required for rehabilitation, and patient quality of life (*National Institute for Health and Care Excellance (NICE), 2017*; *Society of Orthopaedics and Traumatology (SIOT), 2021*; *Bhandari & Swiontkowski, 2017*). For this reason, the percentage of surgical interventions performed within 2 days of hospital admission has become one of the most used health indicators to assess the performance and quality of care. In Italy, specifically, this indicator is used by the *Programma Nazionale Esiti* (National Outcomes Program) to measure and monitor healthcare facilities' performance and standards of care (*Ministero della Salute Agenas, 2019*).

In Europe, more than three quarters (76%) of HF patients aged 65 and over underwent surgery within 48 h of hospital admission in 2017. This proportion was greater than 95% in Denmark and the Netherlands, and around 40% in Latvia and Portugal (*OECD/European Union, 2020*). In Italy, this indicator has progressively improved in recent years, reaching a national average of around 70% in 2017 (*Ministero della Salute Agenas, 2019*).

In late February of 2020, the novel coronavirus disease (COVID-19) (*Gibertoni et al., 2021*; *Lavezzo et al., 2020*) started to spread aggressively around many bordering provinces of the largest and most productive regions of northern Italy: Lombardy, Emilia-Romagna, Veneto, and Piedmont. The first cluster was detected in Lombardy on February 21, 2020—in the following days, the government adopted an increasing number of decrees to limit large social gatherings, closing schools, universities, bars, and restaurants. Following the stay-at-home decree of March 9, 2020, all non-essential business and services were closed, and the entire country was put under lockdown.

The COVID-19 outbreak had a huge impact on the Italian healthcare system: usual treatment pathways were disrupted, and hospitals were reorganized to face this challenge using the limited healthcare resources available (*Placella et al., 2020*). The Italian National Health Service (*Servizio Sanitario Nazionale* (SSN)) struggled to maintain and enhance the surge capacity of services, goods and healthcare workers in order to preserve high standards of care both to COVID-19 and non-COVID-19 patients. To keep COVID-19 and non-COVID-19 patients separated, "Hub and Spoke" models were created (*Regione Emilia Romagna Direzione Generale Cura della Persona Salute e Welfare, 2020*), and COVID-19-dedicated hospitals were set up to isolate contagious patients. Especially during the national lockdown from March 9 to May 4, elective surgery was cancelled, and only trauma, oncologic and urgent surgeries were allowed (*Placella et al., 2020*).

As such, we believe that it is of great importance to examine the impact of the healthcare reorganization adopted to confront the COVID-19 pandemic on the quality of care provided by Italy's SSN for conditions of major public concern such as HF. In this study, we assessed whether the imposition of the national lockdown on March 9, 2020 resulted in a shift in the percentage of patients who received timely surgery for HF compared with that of the pre-lockdown period. Separate analyses were performed on two of the regions of northern Italy most hit by the spread of SARS-CoV-2: Piedmont and Emilia-Romagna, with 4.4 and 4.5 million inhabitants, respectively, as of 2019 (*Golinelli et al., 2017*). As illustrated in Fig. S1, Emilia-Romagna and Piedmont were among the regions of Italy earliest and hardest affected by COVID-19; in particular, the province of Piacenza, Emilia-Romagna, was one of the areas of Italy hardest hit by the pandemic due to its proximity to the epicenter of the first outbreak in Lombardy (*Gibertoni et al., 2021*; *Gatto et al., 2020*; *Rivieccio et al., 2020*).

Identifying similarities and differences in how the two regions faced this unprecedented crisis can be helpful for setting health priorities and identifying entry points to enhance health-system responsiveness (*Williams et al., 2020*).

## MATERIALS & METHODS

We collected the hospital discharge records (HDRs) of all patients admitted to the hospitals of Piedmont and Emilia-Romagna with a principal or secondary diagnosis of hip fracture (ICD-9-CM code 820). In keeping with the specification of the indicator adopted by Italy's National Outcomes Program (*Ministero della Salute Agenas, 2019*), HDRs were excluded from the analysis if at least one of the following criteria was met:

- Non-urgent hospital admission;
- Daytime hospital care, known in Italy as "day hospital admission", which consists in a one-day admission to the hospital without overnight stay to perform diagnostic procedures and/or surgical, therapeutic or rehabilitative care (*Lenzi et al., 2014*);
- Transfer from other hospital;
- Age <65 or >100 years;
- Polytrauma (diagnosis-related group 484–487);
- Diagnosis or medical history of malignant tumors (principal/secondary ICD-9-CM code 140.0–208.9, 238.6, V10);
- Death within 1 day of hospital admission and no surgery to repair HF;
- Admission to a spinal injury unit, rehabilitation hospital or long-term care facility.

Hospitalization rates were obtained as the number of hospital admissions for HF in the resident population aged ≥65 years per 100,000 inhabitants. Population data were retrieved from the Italian National Institute of Statistics (http://demo.istat.it/index_e.html).

Timely HF surgery among the cases described above was defined as any of the following procedures initiated within 2 calendar days after admission to the hospital: closed reduction of fracture without internal fixation (ICD-9-CM codes 79.00, 79.05); closed reduction of fracture with internal fixation (79.10, 79.15); open reduction of fracture without internal fixation (79.20, 79.25); open reduction of fracture with internal fixation (79.30, 79.35);
total or partial hip replacement (81.51, 81.52). We also investigated the percentage of cases surgically treated the next day (day 1) and on the same day as hospital admission (day 0).

Hospital admission rates were standardized by sex and age (<80, 80–84, 85–89, ≥90 years) with direct standardization to Italy's 2020 elderly population. Percentages of surgery were standardized by sex, age and enhanced Charlson index score (0, 1, ≥2) (*Quan et al., 2005*), with direct standardization to the overall population of HFs observed in Piedmont and Emilia-Romagna over the study period.

For descriptive purposes, we also gathered some characteristics of the admitting hospitals; more specifically, we collected hospital type/ownership, hospital location, and average annual caseload of hip fractures.

## Statistical analysis

Owing to the availability of multiple weekly observations in the pre-lockdown and post-lockdown period, we performed an interrupted time-series analysis (ITSA), an example of quasi-experimental design (*Shadish, Cook & Campbell, 2002*). Lockdown-period data were collected from March 11, 2020, to June 9, 2020 (13 weeks, *i.e.,* ≈ 3 months), while pre-lockdown data were collected from December 11, 2019 to March 10, 2020 (13 weeks). To reduce any confounding factors, the same data observed in Piedmont and Emilia-Romagna the year before, *i.e.,* between December 11, 2018 and June 10, 2019 (26 weeks), were included as a control time series with no intervention in the middle of the observation period.

A two-group ITSA regression model can be specified as:

$$Y_t = \beta_0 + \beta_1 T_t + \beta_2 X_t + \beta_3 X_t T_t + \beta_4 Z + \beta_5 Z T_t + \beta_6 Z X_t + \beta_7 Z X_t T_t + \varepsilon_t$$

where $Y_t$ is an aggregated outcome variable measured at each time point $t$, $T_t$ is time since the start of the study, $X_t$ is a dummy variable representing the intervention (pre = 0, post = 1), $Z$ is a dummy variable to denote the cohort assignment ("treatment" or control), and $\varepsilon_t$ is the random error term. Here is the interpretation of the seven parameters that constitute the linear model:

- $\beta_0$ = intercept of the outcome variable in the control group;
- $\beta_1$ = slope of the outcome in the control group until the introduction of the intervention;
- $\beta_2$ = change in the level of the outcome that occurs in the period immediately following the introduction of the intervention in the control group;
- $\beta_3$ = difference between preintervention and postintervention slopes of the outcome in the control group;
- $\beta_4$ = difference in the level between "treatment" and control prior to intervention;
- $\beta_5$ = difference in the slope between "treatment" and control prior to intervention;
- $\beta_6$ = difference-in-differences of the change of level between "treatment" and control;
- $\beta_7$ = difference-in-differences of slopes between "treatment" and control.

As anticipated by the definitions of $\beta_6$ and $\beta_7$, causal inference is provided using the difference-in-differences approach, in which between-period changes in a "treatment"/experimental cohort are compared with changes in a control cohort over

a similar timeframe. The two parameters $\beta_4$ and $\beta_5$ are useful to establish whether the "treatment" and control series are balanced on the level and the trajectory of the outcome variable in the pre-intervention period; if $\beta_4$ and $\beta_5$ are significantly different from 0, conclusions drawn from $\beta_6$ and $\beta_7$ are likely to be biased. A visual exemplification of ITSA is provided in Linden and Adams (*Linden & Adams, 2011*).

We computed heteroskedasticity-robust standard errors, also known as Newey–West standard errors, to make valid inference about the linear regression coefficients. According to the Cumby–Huizinga test (*Cumby & Huizinga, 1992*), there was no evidence of autocorrelation.

In keeping with the specification of the indicator adopted by the Organization for Economic Co-operation and Development, a sensitivity ITSA was performed on HF surgery after excluding HDRs with a diagnosis of hip fracture in secondary position. All analyses were performed using Stata version 15 (Stata Statistical Software: Release 15; StataCorp LP, College Station, TX, USA) (*Linden, 2015*). The significance level was set at 5%, and all tests were two-sided. Code and data used to produce the reported results are made available as Supplemental Files.

Ethical approval to undertake this research was granted from the *Comitato Etico di Area Vasta Emilia Centro* (Submission Number IDECOdE-R (233/2019/0SS/AOUBo)). Access to administrative data was conducted in conformity with the Italian Privacy Code (Legislative decree 5496/2003, amended by Legislative Decree 101/2018), which exempts from the obligation to seek written informed consent when using pseudonymized data that are primarily collected for healthcare management and healthcare quality evaluation and improvement. According to Articles 99–110-bis on medical, biomedical, and epidemiological research (Legislative Decree 101/2018), when investigators use data collected by healthcare systems or previous studies, consulting all the participants would represent a disproportionate effort, considering that safeguards such as key-coding (pseudonymization) are in place to protect the data.

## RESULTS

Hospital admissions for HF in Piedmont and Emilia-Romagna in the control cohort (December 11, 2018, to June 10, 2019) and in the "treatment" cohort (December 11, 2019 to June 9, 2020) are summarized in Table 1. Both regions experienced a drop in the number of hospitalizations in the 13 weeks following the imposition of the first national lockdown as compared to the previous 13 weeks, although the reduction was more pronounced in Piedmont (Piedmont: 148.8 to 121. 6×100,000; Emilia-Romagna: 152.2 to 128. 7×100,000). We registered an increased concentration of admissions to research and university hospitals (Piedmont: 24.0% to 31.4%; Emilia-Romagna: 27.3% to 34.3%), combined with a decrease in the relative number of admissions to LHT and private hospitals (Piedmont: 76.0% to 68.6%; Emilia-Romagna: 72.7% to 65.7%). A summary of patient and hospital characteristics before and after the lockdown is provided in Table S1.

Results of the ITSA on HF hospitalization rates are presented in Table 2 and Fig. 1. In Piedmont, in the second week of March 2020 a strong decrease in weekly hospital

**Table 1  Hospital admissions for hip fracture (× 100,000 inhabitants) in Piedmont and Emilia-Romagna, Italy, by 13-week observation period.**

| Thirteen-week observation period | Piedmont | | | Emilia-Romagna | | |
|---|---|---|---|---|---|---|
| | $n$ | Crude rate | Standardized rate[a] | $n$ | Crude rate | Standardized rate[a] |
| Dec-11-2018 to Mar-11-2019 | 1788 | 161.4 | 159.5 | 1720 | 160.5 | 153.3 |
| Mar-12-2019 to Jun-10-2019 | 1687 | 152.3 | 150.8 | 1731 | 161.6 | 154.6 |
| Dec-11-2019 to Mar-10-2020 | 1698 | 152.2 | 148.8 | 1730 | 160.3 | 152.2 |
| Mar-11-2020 to Jun-09-2020 | 1388 | 124.4 | 121.6 | 1465 | 135.7 | 128.7 |

Notes.

[a]By sex and age with direct standardization to Italy's 2020 elderly population (≥65 years).

**Table 2  Regression table of interrupted time-series analysis on weekly sex- and age-standardized hip-fracture hospitalization rates in Piedmont and Emilia-Romagna before and after Italy's COVID-19 national lockdown.**

| Variable | Piedmont | | Emilia-Romagna | |
|---|---|---|---|---|
| | Coefficient (95% CI) | P-value | Coefficient (95% CI) | P-value |
| Intercept | 12.59 (11.07, 14.12) | <0.001 | 12.99 (12.12, 13.87) | <0.001 |
| $T_t$ | −0.06 (−0.24, 0.13) | 0.546 | −0.17 (−0.29, −0.04) | 0.010 |
| $X_t$ | −0.71 (−2.34, 0.91) | 0.379 | −0.61 (−2.46, 1.24) | 0.510 |
| $X_t T_t$ | 0.12 (−0.14, 0.38) | 0.353 | 0.37 (0.15, 0.58) | 0.001 |
| $Z$ | −0.55 (−2.73, 1.64) | 0.616 | 0.04 (−1.08, 1.15) | 0.949 |
| $ZT_t$ | −0.04 (−0.32, 0.24) | 0.770 | −0.04 (−0.19, 0.12) | 0.627 |
| $ZX_t$ | −2.22 (−4.88, 0.44) | 0.100 | −1.52 (−3.69, 0.65) | 0.164 |
| $ZX_t T_t$ | 0.21 (−0.18, 0.60) | 0.291 | 0.06 (−0.20, 0.32) | 0.629 |

Notes.

Data observed the year before (2018/19) are used for comparison. $T_t$ is time since the start of the study (December 11), $X_t$ is an indicator variable that equals 1 in the weeks 11 to 23 of the tropical year (March 11, 2020/March 12, 2019 to June 9, 2020/June 10, 2019), and $Z$ is an indicator variable that equals 1 in the experimental time series (December 11, 2019 to June 9, 2020). The post-lockdown trend between March 11, 2020 and June 9, 2020 can be obtained as $\beta(T_t) + \beta(ZT_t) + \beta(X_t T_t) + \beta(ZX_t T_t)$.

CI, confidence interval.

admissions for HF was observed as compared to the same week of 2019 ($\beta_6 = [7.95–10.89] − [11.21–11.93] = −2.22 \times 100{,}000$, 95% CI [−4.88–0.44]), although this difference failed to achieve statistical significance (P-value = 0.100). In Emilia-Romagna, this difference-in-differences of change of level was weaker ($\beta_6 = [8.43–10.56] − [10.37–10.98] = −1.52 \times 100{,}000$, 95% CI [−3.69–0.65], P-value = 0.164). In Piedmont, the drop in the number of hospital admissions was followed by a weekly significant increase in the hospitalization rate (+$0.23 \times 100{,}000$, 95% CI [0.03–0.43], P-value = 0.027), although the difference-in-differences of slopes was not significant ($\beta_7 = [0.23+0.10] − [0.06+0.06] = 0.21 \times 100{,}000$, 95% CI [−0.18–0.60], P-value = 0.291). Similarly, in Emilia-Romagna the hospitalization rate increased weekly (+$0.22 \times 100{,}000$, 95% CI [0.11–0.34], P-value <0.001), but the difference-in-differences of slopes was once again not significant ($\beta_7 = [0.22+0.21] − [0.20+0.17] = 0.06 \times 100{,}000$, 95% CI [−0.20–0.32], P-value = 0.629).

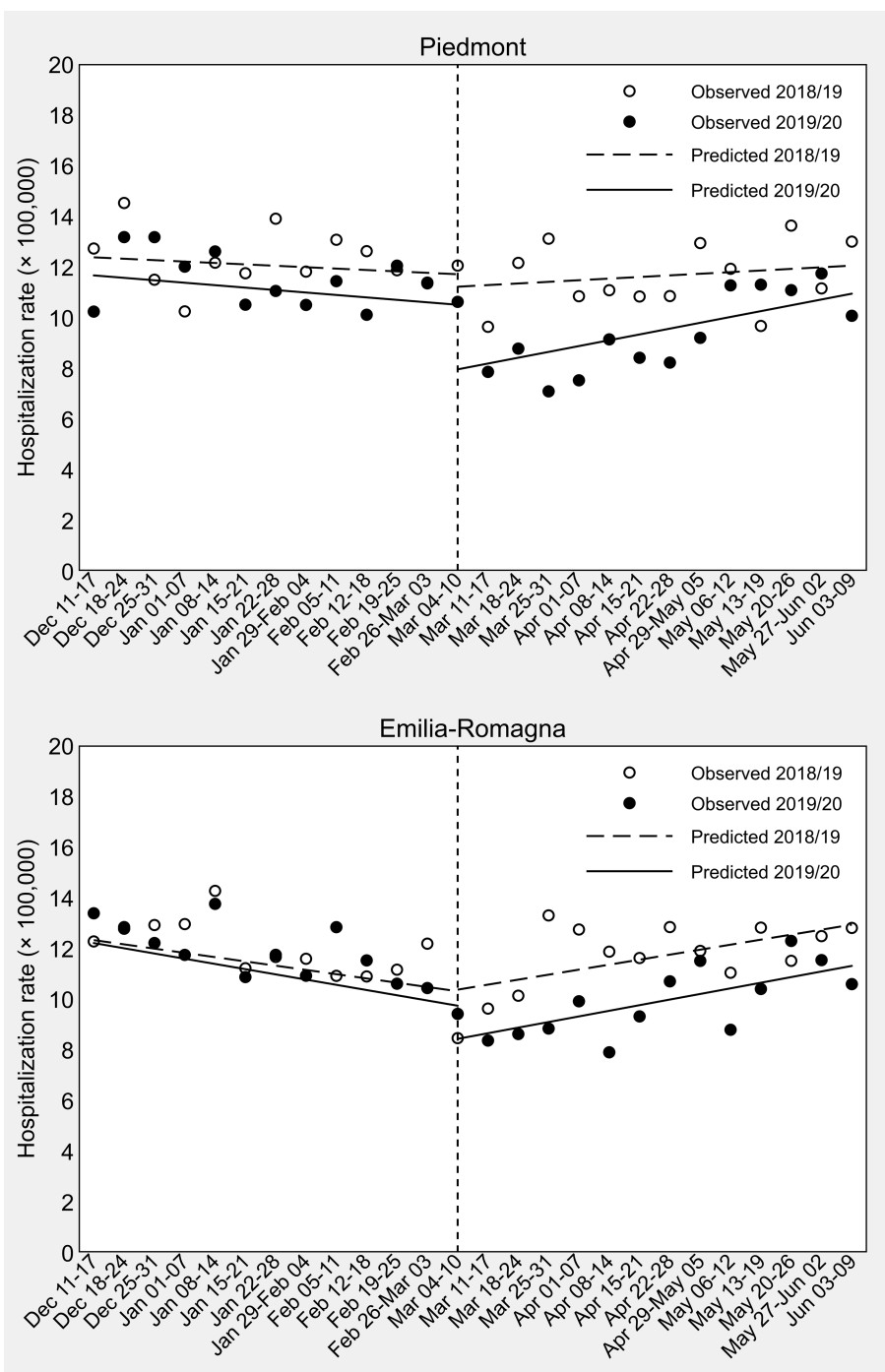

**Figure 1** **Interrupted time-series analysis of weekly sex- and age-standardized hip-fracture hospitalization rates in Piedmont and Emilia-Romagna in the 13 weeks before and after Italy's COVID-19 national lockdown (dashed vertical line).** Data observed the year before (2018/19) are used for comparison. The last day of the control period is June 10, 2019, because 2019 is a common (non-leap) year.

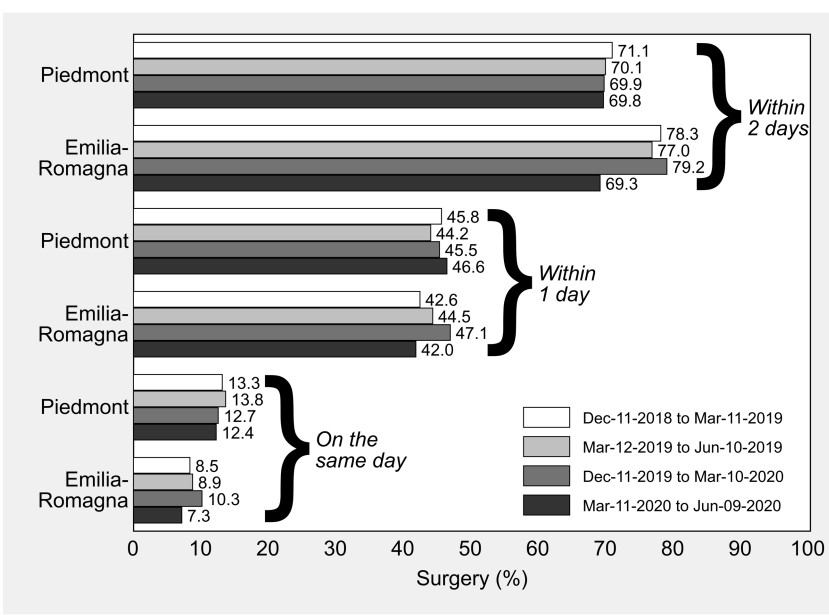

**Figure 2** **Hip-fracture surgery initiated within 2 days, within 1 day and on the same day as hospital admission in Piedmont and Emilia-Romagna, Italy, by 13-week observation period.** All the rates are standardized by sex, age and enhanced Charlson index with direct standardization to the overall composition of hip fractures included in the study.

Percentages of timely HF surgery in Piedmont and Emilia-Romagna in the control cohort (December 11, 2018 to June 10, 2019) and in the "treatment" cohort (December 11, 2019 to June 9, 2020) are illustrated in Fig. 2. In all the study periods preceding the national lockdown, the percentage of surgery initiated within 2 days after hospital admission was higher in Emilia-Romagna than in Piedmont, while between March 11 and June 9, 2020 the standardized percentages of the two regions were similar and just below 70% (Piedmont: 69.8%; Emilia-Romagna: 69.3%).

A visual inspection of the ITSA in Fig. 3 shows that the pattern of change in weekly percentages of surgical care for HF after the imposition of the national lockdown was different in the two study regions. As confirmed by the regression coefficient estimates presented in Table 3, Piedmont did not experience any raise or drop in the amount of surgery from the second week of March 2020. In Emilia-Romagna, on the contrary, after an initial period of stable or even increased timeliness of surgical care, there was a significant decline at a weekly rate of $-1.29\%$ for 2-day surgery (95% CI $[-1.71$ to $-0.88]$, $P$-value $<0.001$), $-2.27\%$ for 1-day surgery (95% CI $[-3.01$ to $-1.54]$, $P$-value $<0.001$), and $-1.07\%$ for same-day surgery (95% CI $[-1.68$ to $-0.45]$, $P$-value $= 0.001$). As shown in Table 3, the difference-in-differences of slopes reached statistical significance for 2-day surgery ($\beta_7 = [-1.29{-}1.28]{-}[-0.19{-}0.59] = -1.79\%$, 95% CI $[-3.37$ to $-0.22]$, $P$-value $= 0.027$), 1-day surgery ($\beta_7 = [-2.27{-}1.61]{-}[0.07{+}0.08] = -4.04\%$, 95% CI $[-5.59$ to $-2.49]$, $P$-value $<0.001$) as well as same-day surgery ($\beta_7 = [-1.07{-}0.59]{-}[0.16{+}0.34] = -2.15\%$, 95% CI $[-3.15$ to $-1.16]$, $P$-value $<0.001$).

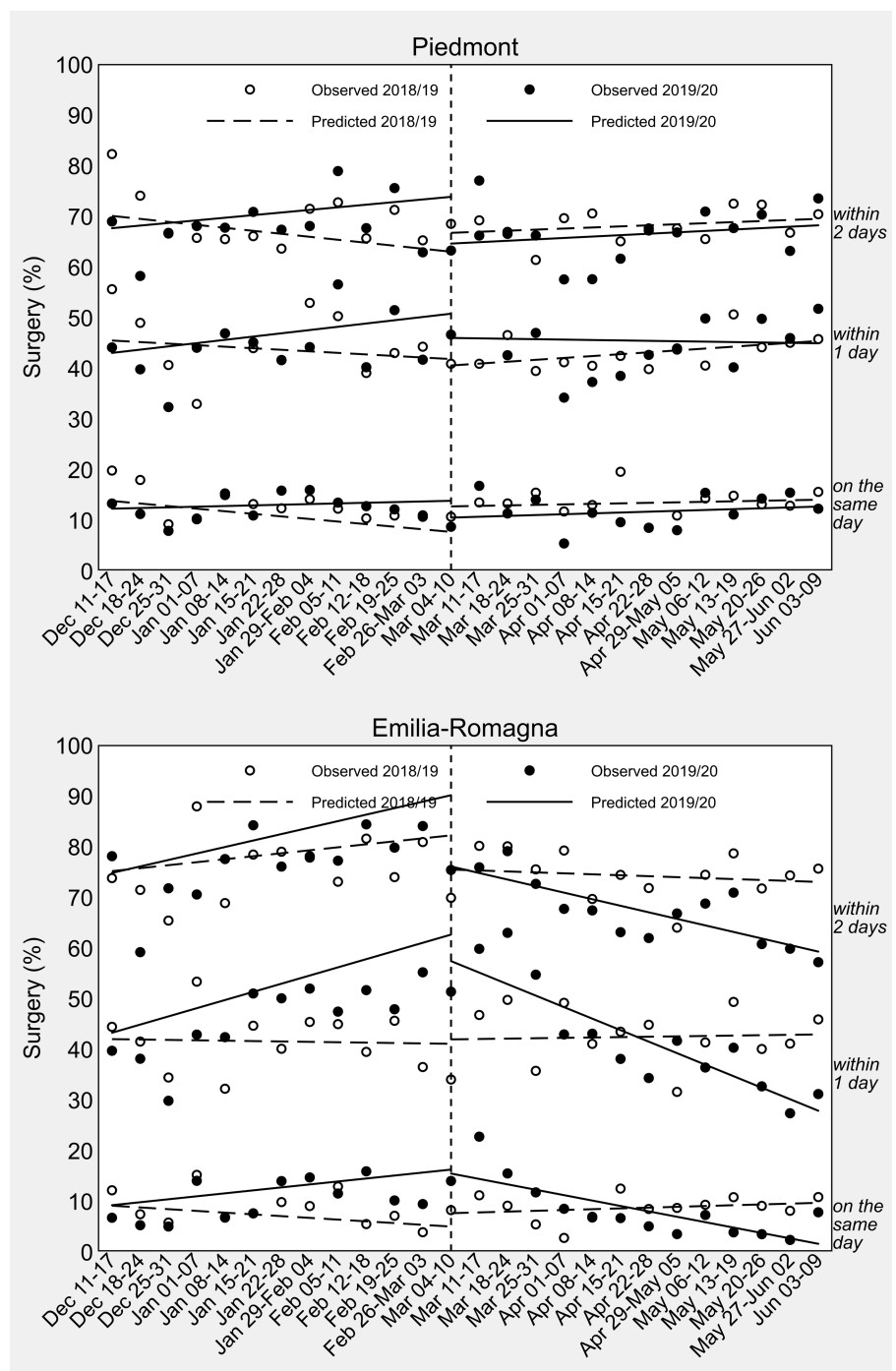

**Figure 3** Interrupted time-series analysis of weekly sex-, age- and comorbidity-standardized percentages of hip-fracture surgery in Piedmont and Emilia-Romagna in the 13 weeks before and after Italy's COVID-19 national lockdown (dashed vertical line). Data observed the year before (2018/19) are used for comparison. The last day of the control period is June 10, 2019, because 2019 is a common (non-leap) year.

**Table 3  Regression table of interrupted time-series analysis on weekly sex-, age- and comorbidity-standardized percentage of surgery for hip fracture in Piedmont and Emilia-Romagna in the 13 weeks before and after Italy's COVID-19 national lockdown.**

| Variable | Piedmont | | Emilia-Romagna | |
|---|---|---|---|---|
| | Coefficient (95% CI) | *P*-value | Coefficient (95% CI) | *P*-value |
| *Surgery within 2 days* | | | | |
| Intercept | 72.36 (64.95, 79.77) | <0.001 | 72.73 (65.55, 79.91) | <0.001 |
| $T_t$ | −0.59 (−1.58, 0.40) | 0.238 | 0.59 (−0.25, 1.43) | 0.166 |
| $X_t$ | 1.37 (−4.99, 7.72) | 0.667 | −4.39 (−11.21, 2.42) | 0.201 |
| $X_t T_t$ | 0.80 (−0.25, 1.86) | 0.133 | −0.78 (−1.80, 0.25) | 0.133 |
| $Z$ | −6.84 (−16.14, 2.45) | 0.145 | −3.11 (−14.56, 8.34) | 0.587 |
| $ZT_t$ | 1.10 (−0.35, 2.56) | 0.133 | 0.69 (−0.71, 2.09) | 0.328 |
| $ZX_t$ | −8.53 (−21.31, 4.26) | 0.186 | −4.56 (−13.73, 4.60) | 0.321 |
| $ZX_t T_t$ | −1.04 (−2.75, 0.67) | 0.227 | −1.79 (−3.37, −0.22) | 0.027 |
| *Surgery within 1 day* | | | | |
| Intercept | 46.62 (38.34, 54.90) | <0.001 | 42.23 (35.33, 49.13) | <0.001 |
| $T_t$ | −0.30 (−1.35, 0.74) | 0.559 | −0.08 (−0.93, 0.78) | 0.858 |
| $X_t$ | −2.49 (−8.94, 3.96) | 0.441 | 0.57 (−8.19, 9.33) | 0.897 |
| $X_t T_t$ | 0.68 (−0.42, 1.77) | 0.220 | 0.15 (−1.02, 1.32) | 0.795 |
| $Z$ | −6.24 (−16.29, 3.80) | 0.217 | −5.50 (−14.36, 3.35) | 0.217 |
| $ZT_t$ | 0.95 (−0.49, 2.39) | 0.192 | 1.69 (0.58, 2.80) | 0.004 |
| $ZX_t$ | 0.28 (−15.01, 15.57) | 0.971 | 0.66 (−11.20, 12.53) | 0.911 |
| $ZX_t T_t$ | −1.40 (−3.34, 0.53) | 0.151 | −4.04 (−5.59, −2.49) | <0.001 |
| *Surgery on the same day* | | | | |
| Intercept | 15.77 (11.47, 20.08) | <0.001 | 10.41 (6.73, 14.10) | <0.001 |
| $T_t$ | −0.51 (−1.03, 0.01) | 0.053 | −0.34 (−0.85, 0.17) | 0.181 |
| $X_t$ | 2.99 (0.01, 5.98) | 0.050 | 1.27 (−3.22, 5.75) | 0.572 |
| $X_t T_t$ | 0.61 (0.05, 1.17) | 0.032 | 0.50 (−0.08, 1.08) | 0.088 |
| $Z$ | −4.08 (−9.17, 1.01) | 0.113 | −3.62 (−8.56, 1.33) | 0.147 |
| $ZT_t$ | 0.64 (0.01, 1.27) | 0.048 | 0.93 (0.19, 1.67) | 0.015 |
| $ZX_t$ | −5.80 (−11.37, −0.22) | 0.042 | 0.30 (−7.51, 8.11) | 0.938 |
| $ZX_t T_t$ | −0.57 (−1.37, 0.22) | 0.154 | −2.15 (−3.15, −1.16) | <0.001 |

**Notes.**

Data observed the year before (2018/19) are used for comparison. $T_t$ is time since the start of the study (December 11), $X_t$ is an indicator variable that equals 1 in the weeks 11 to 23 of the tropical year (March 11, 2020/March 12, 2019 to June 9, 2020/June 10, 2019), and $Z$ is an indicator variable that equals 1 in the experimental time series (December 11, 2019 to June 9, 2020). The post-lockdown trend between March 11, 2020 and June 9, 2020 can be obtained as $\beta(T_t) + \beta(ZT_t) + \beta(X_t T_t) + \beta(ZX_t T_t)$.

Results were virtually unchanged after excluding secondary diagnoses of HF (Table S2 and Fig. S2).

## DISCUSSION

This ITSA was designed to evaluate whether the quality of care received by HF patients changed after the imposition of the COVID-19 national lockdown on early March 2020. Our analysis was restricted to Piedmont and Emilia-Romagna, two large regions of northern Italy severely hit by the pandemic. By identifying similarities and differences in how the

two study regions faced this unprecedented crisis, helpful information can be provided for setting health priorities and identifying entry points for health-system improvement in case of future recurrences of the pandemic, in Italy as in other countries. Furthermore, to the best of our knowledge, there are no published studies that adopted a quasi-experimental design to discern any changes in the quality of HF care during the pandemic.

The most important result of our study is that the percentage of timely surgery for HF remained virtually unchanged in Piedmont after the imposition of the national lockdown, while the Emilia-Romagna system, although performing better than Piedmont before the pandemic, worsened sharply until reaching a proportion below 70% between May and June 2020. This remarkable finding requires an explanation that might be sought in the different capacity and capability of the two regional healthcare systems to respond to the emergency, as well as in the local timing of the epidemic onset.

We acknowledge that the ability to intervene in time on HFs relies both on patients' clinical condition, such as comorbidities or clinical instability management (*e.g.*, coagulation problems) (*Fantini et al., 2011*), and healthcare services' organization (*Fantini et al., 2011*; *Lizaur-Utrilla et al., 2019*; *Sheehan et al., 2017*). However, possible reasons for interregional differences should be attributed to organizational rather than clinical aspects of care; indeed, Piedmont and Emilia-Romagna have slightly different health systems. Like neighboring Lombardy, Piedmont has a hospital-focused system that struggled in the first months to catch up with SARS-CoV-2 infections, due to the lack of a strong primary care system, but that might have ensured high standards of care for non-COVID-19 patients requiring specialized acute settings (*Casula, Terlizzi & Toth, 2020*). Emilia-Romagna relies on a mixed healthcare system, with strong hospital facilities and a well-developed primary care network.

The pandemic hit Emilia-Romagna harder and earlier than Piedmont. Official data report that Piedmont surpassed Emilia-Romagna's hospital burden of COVID-19 on April 10, 2020, about one month after the imposition of the national lockdown, and that reached Emilia-Romagna in terms of cumulative incidence of COVID-19 on April 22, 2020, about 6 weeks after (Fig. S1) (*Gibertoni et al., 2021*; *Gatto et al., 2020*; *Rivieccio et al., 2020*; *Italian Civil Protection Department, 2020*). This may suggest that Piedmont had more time than Emilia-Romagna to organize a proper response to the crisis.

Given these considerations, for the first time since data recording began, the percentage of timely HF surgery in Emilia-Romagna dropped off. This is most likely due to organizational issues related to the emergency, such as the management of testing procedures or the enhancement of the overall capacity of the healthcare facilities. The need for COVID-19 testing procedures before being admitted to the operating room (OR) is an organizational factor that may have increased waiting times. Similarly, waiting for preoperative cardiac tests and other laboratory results may have played a role in delaying surgery (*Lizaur-Utrilla et al., 2019*). A recent study describes how in Lombardy's hospitals, for instance, patients were isolated at admission and sent to a "filtering" ward until the result of nasopharyngeal swabs became available, with a mean response delay, at that time, of 12 to 24 h (*Andreata et al., 2020*). However, our findings cannot be generalized to non-elderly populations and to conditions other than HF, given that a number of treatment pathways succeeded

to maintain high standards of care in Emilia-Romagna. For instance, although overall hospital admissions decreased in a way similar to those for HF, management and outcomes of patients hospitalized with acute myocardial infarction during the same period remained unchanged (*Campo et al., 2021*).

Another aspect to be considered is the workforce surge capacity of healthcare facilities during the pandemic. After the transfer of human resources to sustain intensive care units (ICUs), anesthesiologists may have been under pressure to divide their time between ICUs and ORs, orthopedic surgeons may have been underused or assigned to internal medicine activities in COVID-19 wards, and nurses may have had to adapt to different surgical procedures (*Andreata et al., 2020*). Furthermore, observing COVID-19-related safety protocols is a time-consuming activity that may have created further delays. As a matter of fact, this has an impact on ORs' capacity and readiness, increasing both the "first-case delay", which is an indicator of the delay from the scheduled time for skin incision on the first patient of the day, and the "turnover time", which is the time required for the exit of the patient from the OR, room cleaning, and the entrance of the next patient (*Andreata et al., 2020*).

All these critical aspects might be responsible for the delay in the treatment of patients with HF in Emilia-Romagna after the pandemic outbreak. This highlights the importance of pandemic preparedness and response plans that should include healthcare management issues to respond not only to patients directly affected by the pathogen but also to other patients needing healthcare assistance, such as those with HF or other acute and chronic diseases. These plans should incorporate, for instance, the implementation of specifically trained professionals' networks supported by "health crisis" guidelines to maintain high standards of care even in times of pandemic emergency. Regional healthcare systems should provide healthcare personnel with disaster medicine training courses and reckon on trained "reservists" to be called back in the case of need. Moreover, hospitals and healthcare facilities should schedule simulation trainings and stress tests to speed up peri-surgical activities to be prepared in the event of recurrent crises.

Our findings also show that HF hospital admissions declined in both regions during the first weeks of the lockdown, although the difference-in-differences of change of level was slightly more pronounced in Piedmont than in Emilia-Romagna. This coherent reduction in hospital admissions suggests that the overall impact of the pandemic was similar in the elderly HF populations of the two study regions. This decline has several possible explanations.

Due to the uncontrolled spread of the epidemic across the regions of northern Italy, the central government declared the national lockdown on March 9, 2020. Lifestyle changes, fear of the contagion and sense of civic responsibility (*Rosenbaum, 2020*) may have determined an overall reduction in the number of patients accessing emergency departments (EDs) and hospitals, as confirmed by previous research conducted in Italy and other countries (*Jeffery et al., 2020*; *Santi et al., 2020*). Moreover, by confining people at home, interrupting work activities and reducing road traffic, the frequency of travel- and work-related injuries dropped (*Dolci et al., 2020*); as a consequence, ED visits and hospitalizations for trauma have decreased. Some authors report that domestic accidents

had a relative increase of 94% on total accesses to trauma facilities (*Maniscalco et al., 2020*), determining an increase in the proportion of hospitalizations for HF in the age group ≥65 (*Nuñez et al., 2020*; *Ogliari et al., 2020*) during the first wave of the pandemic. However, in Piedmont and Emilia-Romagna, despite the continued occurrence of domestic accidents, the reduction in overall mobility due to the lockdown led to a decline in the number of HFs and consequent hospitalizations.

Lastly, we found a relative increase in HF hospital admissions to third-level hospitals. We expect this to be the consequence of the reorganization of the healthcare services of both regions. HF and traumas activities were shifted to dedicated hubs, chosen at the regional level among those with more experience and treatment capacity/volumes, in order to maintain high-performance levels even under stressful situations (*Placella et al., 2020*; *Dolci et al., 2020*; *Grassi et al., 2020*). These decisions were made to ensure a rapid increase in the number of ICUs, allowing recruitment as well as replacement for healthcare workers to better assist COVID-19 patients.

### Strengths and limitations

The results of this study should be interpreted considering its strengths and limitations. ITSA is a quasi-experimental research design with a potentially high degree of internal validity, and the addition of a control group (*i.e.*, 2018/19 data) strengthens the causal inference that can be drawn from its results (*Bärnighausen et al., 2017*). By standardizing rates, we also accounted for individual-level confounding differences to evaluate the outcomes of interest at the population level, but ITSA does not allow inferences about the patients that make up the experimental and control cohorts; moreover, our estimates might be affected by some residual confounding due to the impossibility of obtaining relevant information such as patient frailty and clinical complexity from administrative databases. Another limitation to our study is that we did not have access to the hospital reorganization protocols of Piedmont and Emilia-Romagna, so we could not test which one of several potential factors played the leading role in determining our findings. Other limitations are common to all studies based on healthcare administrative data, including lack of accuracy and differences in the coding criteria over time as well as across individuals and institutions. However, there is no reason to believe that such potential source of information bias might have significantly affected our difference-in-differences estimates.

## CONCLUSIONS

In this quasi-experimental study, we found that the COVID-19 pandemic had a similar impact on the HF hospitalization rates of Piedmont and Emilia-Romagna, two of the regions hit earliest and hardest by the virus in Europe, with a coherent relative reduction in HF-related hospitalizations. Conversely, the healthcare services response was different: Piedmont managed to maintain pre-pandemic standards of care, while Emilia-Romagna performed worse despite starting from a better performance level.

Our findings show to what extent the percentage of timely surgery for HF was modified by the pandemic, reflecting local differences in terms of healthcare management, emergency preparedness and response factors. Although there is urgent need for timely and effective

management of COVID-19 patients, it is essential not to forget about other acute and chronic diseases, such as HFs. This draws attention to the enhancement of health services' capacity during emergencies, focusing on the prevention of collateral damage to patients with other diseases, which should be an integral part of any preparedness and response plan aiming to tackle health crises.

### Funding
The authors received no funding for this work.

### Competing Interests
The authors declare there are no competing interests.

### Author Contributions

- Davide Golinelli conceived and designed the experiments, prepared figures and/or tables, and approved the final draft, developed the concept for this study, contributed to the interpretation of the results, and wrote the first draft.
- Jacopo Lenzi conceived and designed the experiments, performed the experiments, analyzed the data, prepared figures and/or tables, and approved the final draft, conducted data analysis, contributed to the interpretation of the results, and wrote the first draft.
- Emanuele Adorno conceived and designed the experiments, analyzed the data, authored or reviewed drafts of the paper, and approved the final draft, contributed to the interpretation of the results, drafted sections of the paper and revised it critically for important intellectual content.
- Maria Michela Gianino and Maria Pia Fantini conceived and designed the experiments, authored or reviewed drafts of the paper, and approved the final draft, contributed to the interpretation of the results, drafted sections of the paper and revised it critically for important intellectual content, equally contributed to this study and are both guarantor for this work.

### Ethics
The following information was supplied relating to ethical approvals (i.e., approving body and any reference numbers):

Ethical approval to undertake this research was granted from the Comitato Etico di Area Vasta Emilia Centro (Submission Number IDECOdE-R (233/2019/0SS/AOUBo)).

### Data Availability
We upload:

+ Supporting data

+ Stata script to perform interrupted time-series analysis.
## Supplemental Information

Supplemental information for this article can be found online at http://dx.doi.org/10.7717/peerj.12046#supplemental-information.

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
