# Peer review of "COVID-19 and regional differences in the timeliness of hip-fracture surgery: an interrupted time-series analysis"

_PeerJ, doi:10.7717/peerj.12046_

## Round 0.1 · original submission · Major Revisions

Besides reviewers' comments, there are several concerns should be addressed before publication.

1. Please edit some English errors (e.g. "a significantly decline").

2. Please modify the equations in a more readable manner. the symbols are sometimes difficult to understand (e.g. "Œ ≤ 0").

3. In Figure captions, there are many "

·

Basic reporting

The paper is well written, interesting and accurate in how resulta are reported. To gain additional readability the paper can be shortened. It is not necessary to describe the two Italian Regions or the origin of the Italian health care system. Some information can be placed in a box but this is not full necessary. It means - you can omit all that info.
The discussion is also lengthy and can be downsized to 800 words. For instance the paragraph bw line 299 and 311 is a repetition of the results without much added value. It can be entirely removed.

Experimental design

The designa is appropriate. In this context there is no need to mention that an ITSA is "a quasi-experimental design that represents a robust alternative to randomized studies when the latter are not feasible". This is not the case in which an ITS is used as a substitute of an RCT.

Validity of the findings

Findings are valid. Confounders are well considered and the stat analysis is appropriate.

Additional comments

A minor suggestion: as identified by the authors the most relevant finding is the dynamic around the percentages of timely surgery for HF. Now this finding is preceded by few other findings that are relatively minor but get too much space.
Wouldn't be better to get this at the centre of the scene ?

Reviewer 2 ·

Basic reporting

No comments

Experimental design

The anlysis is well done.
Regarding the reduction in HP numbers, I can suppose two main causes: 1) lockdown and stay at home campaign hit older people that are more prone to be at home and then the risk of falls is reduced; 2) in the first wave SARS-CoV-2 infection was more frequent in older people and the number of died was higher in older people. Can you discriminate the dominant mechanisms? Other potential explanations?

Validity of the findings

Emilia-Romagna starts from a better performance as compared to Piedmont. It is plausible that before pandemic Emilia-Romagna is treating within 2 days more complex, older, more frial patients as compared to Piedmont. Then, this observation could explain the major drop in Emilia-Romagna as compared to Piedmont because it is more probable that during a pandemic scenario frail patients are at higher risk of under-treatment. Have you data to clarify this point? Do you have other explnations?

Additional comments

I read with great interest the manuscript
It is really important to analyse the performance of regional network during pandemic becasue we can obtain many information to further improve the helthcare
I believe that the Authors have to stress this important issue in their discussion. How can further improved the network?

---

## Round 0.2 · accepted · Accept

I believe all the comments raised by myself and reviewers appropriately then the manuscript was now prepared to be published.

Reviewer 2 ·

Basic reporting

NA

Experimental design

NA

Validity of the findings

NA

Additional comments

All my issues have been addresed